# Pulmonary Hypertension in Left Heart Diseases: Pathophysiology, Hemodynamic Assessment and Therapeutic Management

**DOI:** 10.3390/ijms24129971

**Published:** 2023-06-09

**Authors:** Zied Ltaief, Patrick Yerly, Lucas Liaudet

**Affiliations:** 1Service of Adult Intensive Care Medicine, University Hospital, 1011 Lausanne, Switzerland; zied.ltaief@chuv.ch; 2Service of Cardiology, University Hospital, 1011 Lausanne, Switzerland; patrick.yerly@chuv.ch

**Keywords:** pulmonary hypertension, left heart disease, pathophysiology, therapeutics

## Abstract

Pulmonary hypertension (PH) associated with left heart diseases (PH-LHD), also termed group 2 PH, represents the most common form of PH. It develops through the passive backward transmission of elevated left heart pressures in the setting of heart failure, either with preserved (HFpEF) or reduced (HFrEF) ejection fraction, which increases the pulsatile afterload of the right ventricle (RV) by reducing pulmonary artery (PA) compliance. In a subset of patients, progressive remodeling of the pulmonary circulation resulted in a pre-capillary phenotype of PH, with elevated pulmonary vascular resistance (PVR) further increasing the RV afterload, eventually leading to RV-PA uncoupling and RV failure. The primary therapeutic objective in PH-LHD is to reduce left-sided pressures through the appropriate use of diuretics and guideline-directed medical therapies for heart failure. When pulmonary vascular remodeling is established, targeted therapies aiming to reduce PVR are theoretically appealing. So far, such targeted therapies have mostly failed to show significant positive effects in patients with PH-LHD, in contrast to their proven efficacy in other forms of pre-capillary PH. Whether such therapies may benefit some specific subgroups of patients (HFrEF, HFpEF) with specific hemodynamic phenotypes (post- or pre-capillary PH) and various degrees of RV dysfunction still needs to be addressed.

## 1. Introduction

Pulmonary hypertension (PH) is a hemodynamic condition defined as the occurrence of a mean pulmonary artery pressure (mPAP) > 20 mmHg, assessed by right heart catheterization (RHC) at rest [1]. It can be observed in multiple clinical scenarios which are currently classified in five groups according to common pathobiological pathways, similar clinical features and response to therapy. From a hemodynamic standpoint, pulmonary pressure can increase from the action of three mechanisms: (1) increased cardiac output, (2) increased left atrial pressure (LAP) and (3) constriction and/or remodeling and/or thrombosis and/or embolism of pulmonary arteries (PA). Group 2 PH (PH associated with left heart disease, or PH-LHD) encompasses conditions where elevated LAP is the primary factor of the increased mPAP. Therefore, it includes heart failure with reduced ejection fraction (HFrEF), heart failure with preserved ejection fractions (HFpEF), valvular heart diseases and other congenital/acquired diseases leading to increased LAP [1]. Given the high prevalence of these diseases in contemporary western societies, PH-LHD is by far the most common form of PH and accounts for about 50% of all patients undergoing RHC for PH clarification [2].

### Epidemiology and Significance of PH-LHD in HFrEF and HFpEF

PH-LHD is hemodynamically defined as the occurrence of mPAP > 20 mmHg associated with pulmonary artery wedge pressure (PAWP) >15 mmHg, measured by RHC, as detailed later. If the simultaneously assessed pulmonary vascular resistance (PVR) is ≤ or >2 WU, PH-LHD is further classified as isolated post-capillary PH (Ipc-PH), where PH results from the passive backwards transmission of elevated LAP, or as combined pre- and post-capillary PH (Cpc-PH), which reflects the addition of pulmonary vascular constriction or remodeling to passive PH [1].

The prevalence of PH-LHD is difficult to determine. Indeed, RHC is not routinely performed in HF patients and estimates are mainly based on echocardiographic studies that deliver an indirect assessment of systolic PA pressure with moderate accuracy [3]. In addition, PH definitions have varied over time and studies have been conducted on heterogeneous populations with various degrees of disease severity [4]. Nevertheless, the conjunction of PH with HFrEF or HFpEF has been consistently associated with poor prognosis [5,6], and right ventricular (RV) function is likely the most important prognostic factor in PH-LHD [7,8].

The natural history and significance of PH-LHD are different according to whether it is associated with HFrEF or HFpEF [9]. In HFrEF, PH is likely present at diagnosis, as most patients have signs and symptoms of congestion and hence increased LAP with Ipc-PH. Nevertheless, guideline-directed medical therapy increases cardiac output and lowers filling pressures, which may result in PA pressure normalization [10], particularly with modern therapies such as angiotensin receptor–neprilysin inhibitors and sodium–glucose co-transporter-2 inhibitors [11,12]. A persistently elevated PAP on treatment would therefore indicate more severe HFrEF or concomitant functional mitral regurgitation [9].

At the other end of the disease’s natural history, HFrEF-associated PH-LHD recures in the case of acute decompensation or with the onset of advanced heart failure, indicating intermittent and finally chronic LAP increase that is often associated with reduced cardiac output. With time, some of these patients progress towards Cpc-PH resulting from the progressive pulmonary vasoconstriction and remodeling of their PA and capillary walls [13,14]. Hypothetically, such a reaction may be adaptive by avoiding further increase in the left ventricular (LV) filling pressure. This would limit excessive fluid accumulation in lung interstitial or alveolar spaces and prevent further gas exchange abnormalities and incident hospitalization for acute decompensation [15,16,17]. On the other hand, Cpc-PH imposes a huge supplementary hydraulic load on the RV, resulting in reduced systolic function and increased morbidity and mortality compared to Ipc-PH [18]. In addition, pulmonary veins may also be involved in the remodeling process and correlate better with hemodynamics than PA remodeling [13].

In HFpEF, PH initially occurs only during exercise as a result of a steep increase in LAP with increasing cardiac output [19]. Consequently, RV afterload increases at low workload and an apparently normal RV at echocardiography at rest becomes insufficiently coupled to its hydraulic load (see later) during exercise, which contributes to low cardiac output and limited exercise capacity [20,21]. When present at rest, PH associated with HFpEF mainly reflects evolutive disease with chronically elevated LAP and Ipc-PH. RV systolic dysfunction may already be present, reducing survival probability [9,22]. With time, Cpc-PH may also emerge in HFpEF with a particularly detrimental impact on the RV. Indeed, RV stroke volume reserve becomes very limited with Cpc-PH, and a prominent increase in right atrial pressure can be seen first during exercise and then at rest [23]. Already-enlarged right-sided chambers then further dilate within a restricted pericardial space, resulting in increasing encroachment of the left ventricle (LV), blunted cardiac output and further increase in LV filling pressures [24]. Like with HFrEF, Cpc-PH is associated with reduced survival compared to Ipc-PH in HFpEF [2].

In the present article, we present the current state of knowledge regarding the pathophysiology and hemodynamic assessment of PH-LHD, and provide an updated review of available therapeutic options for its management.

## 2. Pathophysiology of Pulmonary Hypertension in Left Heart Diseases

### 2.1. Passive Upstream Transmission of Elevated Left-Sided Pressure

Thanks to the extremely low resistance to flow in pulmonary arteries (PAs), capillaries and veins, the hydrostatic pressure measured in PAs at the end of diastole (diastolic PA pressure = dPAP) nearly equals that measured in the left atrium (LA) in healthy subjects. As the mitral valve is also opened at that point in the cardiac cycle with no transvalvular pressure gradient between the LA and the LV, normal dPAP is thus mostly predicted by the left ventricular end-diastolic pressure (LVEDP) [25]. The direct hemodynamic consequence of that observation is that blood flow is almost halted at the end of diastole in lungs [26].

If heart failure occurs, filling pressure increases in the LV and is passively transmitted backwards. Indeed, heart failure (HF) is defined by the inability of the heart to match blood flow (i.e., oxygen supply) to the organism’s metabolic demand without activating the Frank–Staling mechanism, hence not increasing its filling pressure [27]. Any aggression or any functional impairment in the LV that would decrease LV stroke volume (SV), LV relaxation or LV compliance is thus followed by increased LVEDP, LAP, pulmonary vein pressure and finally increased dPAP. Numerically, dPAP rises in a ~1:1 mmHg ratio with LAP and LVEDP as a result of the increase in PA blood volume at the end of diastole (Figure 1) [28]. A physiologic gradient ≤ 5 mmHg between dPAP and LAP (or its surrogate, the pulmonary artery wedge pressure, PAWP), the so-called diastolic pressure gradient (DPG) may, however, exist in healthy individuals. It can either arise from the influence of a low positive transmural pressure generated in the airways or because the critical closing pressure of the pulmonary vasculature is reached and PAs collapse [29].

In pulmonary circulation, systolic pressure (sPAP) results from the interaction of the stroke volume (SV) ejected by the right ventricle with the blood volume that already distends PA walls at the end of diastole. Hence, sPAP can be predicted by dPAP and SV with good accuracy (sPAP = 1.41 + (1.61 × dPAP) + (0.09 × SV), and the equation derived to calculate sPAP in healthy subjects predicts sPAP similarly well across many PH conditions including left heart diseases [28]. If SV is kept constant, sPAP increases linearly according to a multiple of dPAP that stands for the decreasing compliance of the more and more distended PA walls with increasing dPAP. Accordingly, sPAP rises in >1:1 mmHg ratio with LAP, which also stands true for the sPAP-dependent mPAP. As a consequence, the pulse pressure (PP, the gradient between sPAP and dPAP) as well as the transpulmonary gradient (TPG, the gradient between mPAP and LAP) progressively increase at higher PAWP and dPAP values for any given fixed SV (Figure 1). On the other hand, larger SVs also produce higher sPAP values at a constant dPAP, but the contribution of SV to sPAP seems less important than the contribution of dPAP [28].

While the trigger of PH-LHD is the passive transmission of elevated left heart pressure, the driving mechanisms of such alterations substantially differ according to the phenotype of heart failure [1]. In the setting of HFrEF (“systolic failure”), progressive LV dilation and eccentric hypertrophy promote a loss of LV chamber compliance, generally resulting from ischemic or dilated cardiomyopathy. These features are associated with secondary mitral regurgitation, subsequent LA enlargement and eccentric remodeling elevating LA pressure. In contrast, in HFpEF (“diastolic failure”), concentric LV hypertrophy and fibrotic changes induce progressive diastolic rigidity with a subsequent increase in LA stiffness and functional mitral regurgitation [31]. These changes develop in the setting of comorbid conditions including diabetes, obesity and hypertension, which also predispose to the development of atrial fibrillation [32], further increasing LA pressure via myocardial oxidative stress and coronary endothelial inflammation [31,33].

### 2.2. Pulmonary Vascular Remodeling and Increased Pulmonary Vascular Resistance

The passive transmission of elevated left-sided pressures to pulmonary circulation determines the most frequent phenotype of PH-LHD, termed Ipc-PH [1]. Over time, the persisting elevation of post-capillary pressure may result in structural modifications in all components of the pulmonary vasculature (veins, capillaries and arteries), a process referred to as pulmonary vascular remodeling [13]. The development of such pulmonary vascular disease (PVD) leads to a progressive increase in pulmonary vascular resistance (PVR), which defines the Cpc-PH phenotype of PH-LHD with significant negative prognostic implications [2].

Increased wall stress in pulmonary capillaries promotes a state of “capillary stress failure”, and breaks in the capillary endothelial and alveolar epithelial layer result in pulmonary edema and alveolar hemorrhages [34]. Fluid and protein accumulation in the interstitium upregulate various inflammatory and pro-oxidant pathways fostering endothelial dysfunction. This results in an imbalance between vasodilation (reduced NO and natriuretic peptide signaling) and vasoconstriction (increased endothelin-1 signaling), elevating vessel resistance to blood flow [16,35,36]. The dysfunctional endothelium adopts a pro-inflammatory and pro-adhesive phenotype, favoring the recruitment of inflammatory cells (primarily monocytes) and the activation of platelets. This results in a progressive amplification of endothelial dysfunction and the further release of inflammatory cytokines (notably IL-6, Interleukin-6), chemokines (notably, MCP-1, monocyte chemoattractant protein-1), growth factors (notably, TGFβ, transforming growth factor beta) and vasoconstricting mediators such as 5-HT (5-hydroxytryptamine) [16,35,36]. Interactions of these various cascades elicit the remodeling of the extracellular matrix and proliferation of fibroblasts/myofibroblasts [31]. The consequent thickening of the alveolar septa impairs gas exchange, which is reflected by reduced gas diffusion capacity and increased dead space ventilation in patients with heart failure and a Cpc-PH phenotype [24]. Furthermore, intimal thickening and the unrestrained proliferation of vascular smooth muscle with progressive medial hypertrophy concur to narrow the lumen of pulmonary vessels and further increase PVR [31]. These alterations affect both the venous and the arterial segments, as precisely documented in a major histopathological study by Fayyaz et al. [13]. Interestingly, these authors found that the severity of PH-LHD correlated more strongly with venous than arterial thickening, which is similar to the pattern observed in PH secondary to pulmonary veno-occlusive disease. Several extensive reviews dealing with the pathobiology of pulmonary vascular remodeling have been published recently [35,36,37].

Only a minority of patients with heart failure develop significant PVD and a pre-capillary phenotype, implying that some specific factors may contribute to such alterations. A genetic predisposition has been highlighted in a study comparing the genetic profile of patients with Cpc-PH and Ipc-PH. The authors identified 141 differentially regulated genes in Cpc-PH which were enriched in pathways involving the extracellular matrix, cell structure and immune function [38]. Metabolic factors represent additional contributors to the development of PVD in patients with heart failure. Obesity and diabetes are associated with increased inflammatory signaling and oxidative stress, which may contribute to vascular remodeling in the pulmonary circulation [39,40,41]. Importantly, such metabolic disorders represent common comorbidities associated with HFpEF, which may underly the more frequent development of a Cpc-PH phenotype in HFpEF (up to 38%) than in HFrEF (~18%) patients [31].

## 3. Hemodynamic Assessment of PH-LHD

### 3.1. Hemodynamic Phenotypes of PH-LHD

The invasive assessment of pulmonary hemodynamics using right heart catheterization (RHC) in patients with suspected PH-LHD allows practitioners to confirm the diagnosis, informs them about the underlying mechanisms of elevated PA pressure and provides data relevant for its management and prognostication [1,42]. The most recent definition of PH-LHD includes a mPAP > 20 mmHg together with a PAWP > 15 mmHg, reflecting the passive increase in mPAP due to the proportionate backward transmission of high LA pressure. When associated with normal PVR, these changes determine the Ipc-PH phenotype of PH-LHD, as previously mentioned (Table 1) [1]. A strict methodology for PAWP measurement is mandatory for accurate diagnosis. The current recommendation is to take the value at the nadir between a and c waves of the PAWP trace, or at ~150 ms after the onset of the Q wave with QRS-gated measurements. In the presence of significant v waves, PAWP should be averaged over the entire cardiac cycle. The value should be systematically obtained at the end expiration [43].

In the subset of patients developing PVD, the increase in mPAP becomes out of proportion to the increased PAWP and is associated with an increase in PVR, calculated from the ratio of the transpulmonary gradient (PAPm-PAWP) to the cardiac output. This defines the Cpc-PH phenotype of PH-LHD [1]. The cut-off value of PVR differentiating Ipc-PH from Cpc-PH was 3 Wood Units (WU) [42], but this cut-off was reduced to 2 WU in the most recent guidelines on PH [1]. This change was based on the results of a large retrospective cohort study of more than 40,000 patients undergoing RHC, which indicated that a PVR of 2.2 WU or more was associated with a sizeable increase in adjusted mortality risk in patients with elevated pulmonary artery pressure [44]. A graphic algorithm for the hemodynamic diagnosis of PH-LHD is provided in Figure 2.

The development of PVD and increased PVR in the setting of left heart disease is associated with increased right ventricular (RV) dysfunction and mortality regardless of the etiology (HFrEF, HFpEF or valvular diseases) [2,18,45,46]. This association of PVR with outcome has been highlighted in two recent meta-analyses reporting a 7–9% increase in the risk of adverse outcomes for each unitary increase in PVR in patients with PH-LHD [47,48]. Besides PVR, a value of TPG greater than 12 mmHg was previously considered as a surrogate marker of an out-of-proportion increase in mPAP with respect to PAWP in patients with PH-LHD [49]. However, as mentioned above, the TPG is sensitive to changes in PAWP and also to the recruitment and distension of the pulmonary vessels [30], and is therefore not relevant to detect vascular remodeling in patients with PH-LHD. The value of TPG has thus been abandoned to differentiate Ipc-PH from Cpc-PH [50]. Another potential surrogate indicator of pulmonary vascular pathological dysfunction is the diastolic pressure gradient (DPG) [51], which is normally < 5 mmHg, as discussed previously. Accordingly, a value of DPG ≥ 7 mmHg had been previously incorporated in the diagnostic criteria of Cpc-PH [52,53]. Nevertheless, DPG is a small value prone to measurement errors [51] and, due to uncertainties regarding its specificity and real prognostic implication [54], it has been removed from the most recent definitions of Cpc-PH [1]. 

### 3.2. Provocative Testing

Provocative testing refers to the use of RHC during either exercise or fluid challenge (Table 2). It is mainly indicated to detect exercise PH in patients with a clinical probability of HFpEF, but who display normal resting hemodynamics (mPAP < 20 mmHg with a PAWP < 15 mmHg) [1,42]. Exercise stress testing must be dynamic in nature (e.g., cycling), but not static (e.g., handgrip) in order to produce a meaningful increase in cardiac output (CO) which promotes an augmentation in pulmonary blood flow associated with an increase in mPAP [55]. The degree of mPAP increase in response to the change in CO is computed as the slope of the mPAP/CO ratio during a stepwise workload increment (three–five steps with hemodynamic measurements at each step) [55,56]. The normal reference values for mPAP/CO are 0.5–3 mm Hg/L/min, or a maximal value of mPAP of 30 mmHg at a CO of 10 L/min, corresponding to a maximal value of total PVR (mPAP/CO) of 3 WU [57]. Values above this normal range define exercise-induced PH [55,56].

An increased slope of mPAP/CO may depend either on an increased PVR or an increased PAWP. The differential diagnosis therefore relies on the determination of PAWP during exercise [57]. A linear increase in PAWP normally occurs during exercise, resulting from the increased filling pressure of the LV required to augment CO, in accordance with Starling’s law of the heart. It has been thus reported that PAWP can increase up to 25 mmHg in trained athletes during exercise [58]. In patients with HFpEF, this exercise-induced increase in PAWP is much greater, due to the impaired diastolic relaxation and reduced compliance of the LV, promoting a steep increase in left atrial pressure even during moderate exercise [59,60]. A threshold value of PAWP ≥ 25 mmHg is currently recommended for the diagnosis of HFpEF [60]. However, it seems more appropriate to compute the ratio of PAWP/CO change during incremental exercise, as this method appears more sensitive to refine early HFpEF diagnosis, with an upper normal limit of 2 mmHg/L/min [19]. Therefore, criteria to diagnose occult PH-LHD during exercise incorporate an mPAP/CO > 3 mmHg/L/min together with a PAWP/CO > 2 mmHg/L/min [1,55].

Owing to technical and practical challenges associated with exercise hemodynamic assessment, an alternative provocative test to detect occult PH-LHD is the fluid challenge test [61]. The current recommendation is to administer a 7 mL/kg bolus of normal saline over 5 min and to measure PAWP at the end of infusion [62]. Under normal conditions, PAWP should not exceed 18 mmHg; therefore, a positive fluid challenge detecting occult PH-LHD is defined as a PAWP > 18 mmHg [42].

## 4. The Prognostic Importance of Pulmonary Artery Compliance in PH-LHD

The higher mortality associated with an increased PVR in Cpc-PH reflects the progressive development of RV dysfunction in the presence of an elevated afterload [24]. It must be emphasized that PVR only represents the static component of RV afterload, but it does not consider the contribution of the pulsatile component of RV afterload [63]. The latter is captured by the pulmonary artery compliance (PAC), which defines the relationship between the volume increase and the corresponding change in pressure of the PA during systole. In the clinical setting, PAC is usually calculated from the ratio of the stroke volume (SV) to the PA pulse pressure (PP = PA systolic minus PA diastolic pressure), although this calculation overestimates the true PAC, as it does not consider the fraction of SV flowing towards the periphery during systole [64]. The currently defined normal reference value for PAC is >2.3 mL/mmHg [1].

An important feature of the pulmonary circulation is that PVR and PAC are inversely related according to a hyperbolic relationship (Figure 3A), and that their product, termed the arterial time constant (RC time), remains invariable (~0.5 s) regardless of the etiology and severity of PH [65,66]. This peculiarity is related to the fact that, at variance with the systemic circulation, PAC depends largely on the distal vasculature, while the proximal large pulmonary arteries contribute only ~20% of the total PAC [67]. In addition, due to the nonlinear relationship between vessel diameter and compliance, an elevated arterial pressure secondary to increased PVR results in stiffer arteries with reduced compliance [66,68]. This inverse hyperbolic relationship indicates that, at an early stage of pulmonary vasculopathy, the reduction in PAC is much greater than the increase in PVR, whereas the opposite is true at an advanced stage of the disease [64]. The situation is notably different in patients with LHD, as the passive backward transmission of elevated left atrial pressure produces a certain degree of stiffening in the pulmonary circulation, resulting in a reduction in the pulmonary RC time and in PAC [69,70]. Therefore, an increase in LV filling pressure produces, de facto, a greater reduction in PAC at any level of PVR, thereby augmenting the pulsatile afterload of the right ventricle [69].

The importance of a reduced PAC in promoting RV dysfunction and worsening prognosis in patients with PH-LHD has been reported by several investigators. Al-Naamani et al. identified that a PAC < 1.1 mL/mmHg was 91% sensitive in predicting mortality in patients with PH-LHD associated with HFpEF, with better discriminatory ability than PVR [71]. In a large population of patients with advanced HFrEF, Dupont et al. reported that PAC, dichotomized as > or <2.5 mL/mmHg, was a stronger predictor of all-cause mortality or transplantation than PVR [68]. In a comparable population of HFrEF patients, Pellegrini et al. identified a cut-off value of PAC < 2.15 mL/mmHg that was associated with poor prognosis, independently from the type of PH-LHD (Ipc-PH or Cpc-PH) [72]. Furthermore, extracting data from three large databases of patients with HFrEF and HFpEF, Tampakakis et al. reported that PAC more consistently predicted mortality than PVR and remained predictive of poor outcomes in the subgroup of patients with normal PVR (Ipc-PH) [73]. To sum up current evidence on this issue, a recent meta-analysis including 9600 patients with PH-LHD indicated that PAC was associated with an increased risk of adverse outcomes with a hazard ratio of 0.76 (95% CI 0.69–0.84) [47]. Importantly, PAC also represents a critical determinant of outcome in patients presenting with acute left ventricular dysfunction, as shown by our group in a cohort of 91 patients with cardiogenic shock. In this study, PAC was the only hemodynamic variable significantly associated with mortality in univariate and multivariate analyses [74].

**Figure 3 ijms-24-09971-f003:**
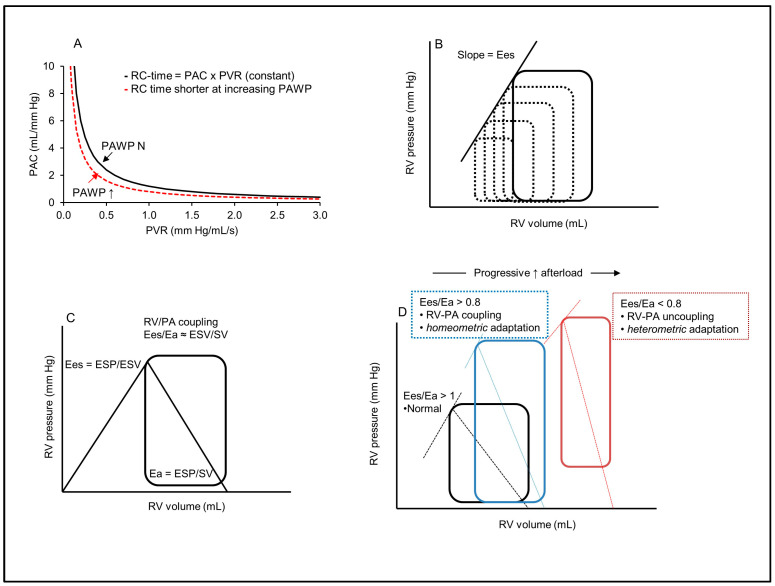
Right ventricle afterload, contractility and the concept of right ventricle–pulmonary artery coupling. (**A**) Pulmonary vascular resistance (PVR) and compliance (PAC) are related according to a hyperbolic relationship, and their product, the pulmonary artery time constant (RC-time), is invariable. In left heart disease, the increased PAWP shifts the relationship to the left, with a reduction in the RC-time, and more reduced PAC at any value of PVR (increased pulsatile afterload). (**B**) Right ventricle pressure–volume (PV) curves for the determination of end-systolic elastance (Ees). Ees is expressed as the slope of the end-systolic pressure–volume relationship of successive PV loops at rapidly reduced preload. (**C**) Isolated right ventricle PV curve for the simplified assessment of Ees (end-systolic pressure/end-systolic volume, ESP/ESV) and effective arterial elastance (Ea) as a measure of RV afterload (Ea = ESP/SV). The ratio of Ees to Ea defines RV-PA coupling. (**D**) Right ventricle PV loops with different Ees/Ea ratios. The blue loop indicates an RV exposed to increased afterload, with adapted contractility through homeometric adaptation and maintenance of RV-PA coupling. The red loop shows an uncoupled RV with increased dimension (heterometric adaptation). Adapted from Refs. [64,75,76].

## 5. The Right Ventricle in PH-LHD

RV dysfunction occurs frequently in patients with left heart diseases and is associated with significant negative prognostic implications, both in HFrEF and HFpEF patients [77,78]. In an echocardiographic study performed in 657 age- and gender-matched groups of patients with HFpEF, HFrEF and controls without HF, Bosch et al. reported a prevalence of RV systolic dysfunction of 30–40% in HFpEF and ≥60% in HFrEF patients [7]. In this study, RV dysfunction was assessed by a tricuspid annular plane systolic excursion (TAPSE) < 17 mm and an RV longitudinal strain (RVLS) < −20%. The increased RV afterload due to reduced PA compliance (Ipc-PH) and elevated PVR (Cpc-PH), as well as intrinsic RV myocardial abnormalities, concur with precipitate RV dysfunction in this setting [1]. This results from impaired functional interactions between the RV and the pulmonary circulation defining the cardiopulmonary unit [64].

### 5.1. Physiology of the Coupling between the Right Ventricle and the Pulmonary Circulation

The maintenance of normal RV function requires its contractile adaptation to the load imposed by the pulmonary circulation for optimal energy transfer from the RV to the PA. This adaptation defines the concept of RV–afterload *matching* [79] or RV-PA *coupling* [64]. RV contractility depends on the contractile force of the myocytes (inotropy) and RV muscle mass. It also depends on the LV, which contributes 20–40% of the RV pressure generation, via systolic interdependence related to the shared septum and common circumferential myocardial fibers and is, therefore, negatively impacted by a depressed LV systolic function, as in HFrEF [80]. RV contractility is best described by its end-systolic elastance (*Ees*), which reflects the maximal pressure that the RV can generate at end-systolic volume [64]. Ees is generally assessed by the slope of a line connecting end-systolic pressures and volumes from a family of pressure–volume (PV) loops obtained at rapidly reduced preload [64] (Figure 2B). A single-beat method to compute Ees from a single PV loop and RV pressure trace has also been described [81]. From a practical standpoint, Ees can be estimated by the ratio of end-systolic pressure and volume (Ees = ESP/ESV) [75] (Figure 2C).

RV afterload represents the sum of all forces opposing blood flow within the pulmonary circulation, which depends on the interactions between PVR (steady afterload), PAC (pulsatile afterload) and pulse wave reflections in the pulmonary arteries [75]. RV afterload is best described by the computation of pulmonary vascular impedance through the spectral analysis of synchronized pressure and flow waves within the proximal PA [82]. However, it is more commonly expressed as a lumped parameter termed the *effective arterial elastance* (Ea). The latter corresponds to the slope of the relationship between RV end-systolic pressure (ESP) and stroke volume (SV) on an RV PV curve (Ea = ESP/SV) (Figure 2C) [83]. For practical purposes, the value of mPAP may be used as a surrogate of RV ESP, using the following correction: ESP (mmHg) = (1.65 × mPAP) − 7.79 [84].

RV-PA coupling is then determined by the ratio of their respective elastance, Ees/Ea, which should remain > 1 (normally in the range of 1.5 to 2) for optimal RV output at minimal energy cost [85]. Thus, under conditions of elevated afterload, RV-PA coupling is maintained through an increased RV systolic function, with preserved stroke volume and filling pressure [64]. This adaptation depends both on a reflex increase in RV contractility (the so-called Anrep effect), and on progressive RV muscle hypertrophy, with preserved RV dimensions (*homeometric* adaptation). Failure of such adaptation results in RV-PA uncoupling, with progressive RV dilation and increased filling pressure (*heterometric* adaptation), leading to RV failure and upstream congestion [86] (Figure 2D). Moreover, RV dilation reduces LV filling through leftward shift of the interventricular septum (diastolic interdependence), which may result in depressed LV systolic function, reduced cardiac output and further increase in LV filling pressure [80].

Clinically, RV-PA coupling can be estimated from a simplified version of the Ees/Ea equation using only volumes, determined by cardiac magnetic resonance imaging: Ees/Ea = (ESP/ESV)/(ESP/SV) = SV/ESV [85] (Figure 2C). Although this method underestimates the true value of Ees/Ea, its prognostic significance has been clearly established [64], with a cut-off value < 54% determining the onset of RV failure with a significant reduction in survival [75,87]. The RV ejection fraction (EF) can also be used to compute SV/ESV (given that SV = EDV − ESV), such as SV/ESV = EF/(1 − EF), with a cut-off value of RV EF < 35% defining RV-PA uncoupling [75]. Finally, RV-PA coupling can be indirectly evaluated using echocardiographic assessment. Using this method, the value of TAPSE (an index of longitudinal RV fiber shortening) is obtained as a surrogate for RV contractility (Ees) and the value of systolic PAP (PAPS) as a surrogate for RV afterload (Ea). The ratio TAPSE/PAPS is then computed to estimate Ees/Ea, and a cut-off value < 0.31 mm/mmHg has been proposed to discriminate RV-PA uncoupling [88]. Table 3 provides a summary of the relevant variables for the determination of RV-PA coupling.

### 5.2. RV-PA Uncoupling in PH-LHD

Several investigations have emphasized the critical importance of RV-PA uncoupling in the setting of left heart disease. In 293 patients with PH-LHD, Guazzi et al. reported that the TAPSE/PAPS ratio, as an index of RV-PA coupling, was the strongest independent predictor of disease severity (NYHA functional class) and mortality, with an optimal dichotomous threshold of 0.36 mm/mmHg, regardless of LVEF (HFrEF or HFpEF) [89]. The same authors confirmed these observations in a later study exploring RV-PA coupling in a cohort of 387 patients with HFpEF. Patients with a TAPSE/PAPS ratio < 0.35 mm/mmHg displayed higher plasma levels of natriuretic peptides, reduced exercise capacity and a significantly worse outcome [90]. Furthermore, Gerges et al. showed that PH-LHD patients with Cpc-PH had significantly worse RV-PA coupling and shorter survival than patients with Ipc-PH. In this study, the cut-off values of TAPSE/PAPS discriminating Cpc-PH from Ipc-PH was 0.31 mm/mmHg in HFpEF and 0.27 mm/mmHg in HFrEF patients [91]. RV-PA uncoupling has also been highlighted as an important mechanism of exercise intolerance in patients with PH-LHD related to HFpEF. Borlaug et al. found that such patients failed to adapt their RV systolic function (impaired contractile reserve) to the increase in pulsatile afterload occurring during exercise, resulting in RV-PA uncoupling, elevated filling pressures and insufficient cardiac output increase relative to their metabolic needs [20]. This dynamic impairment of RV contractile reserve was further amplified in patients with an associated pre-capillary component (Cpc-PH), who developed marked RV-PA uncoupling associated with significantly reduced peak O_2_ consumption during exercise [24,63].

While the above data suggest that both an increased afterload and an intrinsic RV contractile impairment contribute to RV-PA uncoupling in the setting of HFpEF, a recent study by Oakland et al. has challenged this hypothesis. In a mixed population of PAH and PH-LHD in HFpEF patients, these authors determined RV afterload by computing a PA wave reflection coefficient and assessed RV-PA coupling using the Ees/Ea ratio. Both PAH and HFpEF displayed a comparable increase in RV afterload and reduction in Ees/Ea. However, in marked contrast with PAH, in which RV-PA uncoupling was strictly related to RV afterload, RV dysfunction and RV-PA uncoupling in HFpEF patients was independent from afterload. Therefore, these data support that an inherent abnormality in the RV myocardium represents the primary mechanism of RV-PA uncoupling in this population [92]. These data may provide an explanation for the limited effects of therapeutic strategies aiming to reduce afterload in patients with PH-LHD, as discussed below.

## 6. Therapeutic Management of PH-LHD

As discussed in the previous sections, the primary driver of PH-LHD is the upstream transmission of the elevated LA pressure to the pulmonary circulation, promoting an increase in the pulsatile afterload of the RV by reducing PA compliance (Ipc-PH). In a subset of patients, chronic PAWP elevation results in progressive PVD, with the subsequent elevation in PVR leading to a further rise in RV afterload (Cpc-PH). Ultimately, the chronically overloaded RV becomes uncoupled from the pulmonary circulation, an evolution further precipitated by the lack of RV contractile reserve in the context of heart failure (RV-PA uncoupling). Therefore, therapy for PH-LHD should, first, aim to decrease PAWP to improve PA compliance and prevent PVD and, second, to reduce PVR in the case of established PVD, with the final goal to prevent or treat RV-PA uncoupling.

The initial trigger (elevated left-sided pressure) and the hemodynamic phenotypes of PH-LHD are comparable regardless of the cause of heart failure (HFpEF, HFrEF). However, there are important differences in terms of comorbidities and in the pathophysiological mechanisms responsible for the increase in left-sided filling pressures. Additionally, there may be some dissimilarities in the molecular mechanisms of pulmonary vascular remodeling, as previously mentioned [39]. Therefore, PH-LHD encompasses several entities, which must be considered when addressing the effects of therapeutics [31]. For these reasons, the various treatment approaches detailed in the following sections will be presented in the perspective of the underlying etiology of PH-LHD.

### 6.1. Guideline-Directed Medical Therapies for Heart Failure

The obvious first step in treating PH-LHD is to optimize the management of the underlying cardiac disease [1]. This must include the appropriate use of guideline-directed medical therapies (GDMTs), together with the treatment of comorbidities notably associated with HFpEF, primarily, diabetes, obesity and hypertension. Updated guidelines for the therapy of HFrEF and HFpEF have been recently published [93]. The benefit of such optimization has been particularly emphasized in the CHAMPION study. The study evaluated a strategy of therapeutic adaptation based on the home transmission of PAP measurement as a surrogate of PAWP using an implantable sensor (CardioMEMS^TM^) in patients with symptomatic heart failure (irrespective of EF) and PH-LHD [94]. The intervention group showed a 33% reduced rate in hospital admissions for heart failure and a lower composite outcome of death and all-cause hospital admissions over a 31-month follow-up. This was associated with a significantly increased use of heart failure medications, most notably diuretics. Comparable results have been obtained in subsequent clinical studies using the same strategy of PAP-based management of HF medication [95,96], which underscored the major impact of diuretic adaptation to control PAP in HF patients.

Recent data also indicated a potential impact of novel drugs used in HF therapy in reducing PAP in patients with PH-LHD, including angiotensin receptor–neprilysin inhibitor (ARNI) and sodium–glucose cotransporter-2 (SGLT-2) inhibitors [97]. The Paradigm-HF trial showed that a combination of valsartan with sacubitril, an inhibitor of neprilysin, improved outcomes in patients with HFrEF [98]. Since neprilysin degrades natriuretic peptides (NP), which possess inherent pulmonary vasodilating properties [99], treatment with sacubitril could theoretically enhance such vasodilation. In animal models of PH, treatment with sacubitril reduced RV systolic pressure, improved RV-PA coupling and reversed RV remodeling [100,101]. In humans, a retrospective study on 18 HFrEF patients bearing an implanted CardioMEMS^TM^ PAP sensor reported that the transition from an angiotensin-converting enzyme (ACE) inhibitor or an angiotensin receptor blocker (ARB) to sacubitril/valsartan resulted in a significant reduction in PAP. The fall in PAP was hyperacute (<48 h) in a majority of patients and occurred regardless of normal or elevated PVR [102]. Comparable results have been described in small observational studies, some also reporting improved RV-PA coupling and RV function following the introduction of sacubitril/valsartan, as recently reviewed by Zhang et al. [12]. Randomized control trials (RCT) are needed to confirm these findings and to investigate the impact on clinical outcomes and mortality.

SGLT-2 inhibitors (Gliflozins), which are effective in the treatment of type 2 diabetes, have recently been associated with clinical benefits in patients with HFrEF and HFpEF, and have been incorporated in the GDMTs (HFrEF, 1a recommendation; HFpEF, 2a recommendation) [93]. Among the multiple possible actions of SGLT-2 inhibitors, natriuresis and improved endothelial function could contribute to improve pulmonary hemodynamics in HF patients. This hypothesis was recently addressed in the EMBRACE-HF trial, which randomized 65 HF patients (both HFpEF and HFrEF) with an implanted CardioMEMS sensor to receive either empagliflozin or a placebo [11]. Results indicated that empagliflozin significantly reduced PAP, an effect starting after 1 week that progressively amplified over time, which appeared independent of diuretic management. In another randomized trial performed in patients with type 2 diabetes and exercise-induced PH (occult PH-LHD), treatment with dapagliflozin vs placebo blunted the increase in RV systolic pressure and LV filling pressure evaluated by echocardiography upon repeated exercise testing after 6 months [103]. These promising findings suggest that SGLT-2 inhibitors could offer novel opportunities to control PA pressure in heart failure patients, as emphasized in a recent review on this topic [97].

### 6.2. Additional Therapies

#### 6.2.1. Levosimendan

Levosimendan (Levo) is a calcium sensitizer, potassium ATP (K_ATP_) channel activator and phosphodiesterase-3 inhibitor [104] with inotropic, vasodilatory and cardioproptective activity. It is widely used (outside of the US) in the treatment of acute and advanced heart failure [105]. Experimental data showed that Levo could attenuate pulmonary vascular remodeling in an animal model of PH, attributed to antiproliferative and anti-inflammatory effects mediated by K_ATP_ channel activation [106]. Moreover, beneficial actions of Levo on adverse RV remodeling in experimental PH have been reported [107,108]. In humans, some limited evidence suggests that Levo may reduce PAP and PVR and improve RV function in patients with PAH [109,110]. A review of the effects of Levo on pulmonary circulation has been recently published [111].

In PH-LHD, several studies indicated beneficial effects of Levo on pulmonary hemodynamics and RV function in patients with HFrEF (Table 4). Slawsky et al. randomly assigned 146 HFrEF patients (mean EF 21%) either placebo or Levo (0.1–0.4 μg/kg/min for 6 h) and found a significant reduction in mPAP and PVR in the Levo group [112]. Parissis et al. prospectively explored the effects of Levo (0.1–0.2 μg/kg/min for 24 h) compared to placebo in 54 patients with advanced HFrEF (mean EF 21%). After 48 h, patients treated with Levo displayed significant reductions in systolic PAP, improved echocardiographic indices of RV systolic and diastolic function, and decreased plasma levels of natriuretic peptides [113]. The positive effects of Levo in reducing PAP and improving RV function in HFrEF have been recently confirmed in a meta-analysis of eight studies totalizing 390 patients [114]. This meta-analysis, however, did not indicate consistent effects of Levo in decreasing PVR.

The effects of Levo have also been evaluated in HFpEF patients in the phase II HELP Study, which assessed the effects of repeated, home-delivered, intravenous infusions of Levo (0.075 μg/kg/min for 24 h once a week over 6 weeks) [115]. The primary outcome was a reduction in the increase in PAWP upon a 25 W exercise challenge, and the secondary endpoint was an improvement in 6 min walk distance (6MWD). Although Levo was associated with a reduction in resting PAWP, there was no significant difference in the primary outcome. In contrast, patients receiving Levo displayed a significant improvement in 6MWD (29 m), which is comparable to that reported for approved pulmonary vasodilators for PH [115]. Interestingly, there were no effects of Levo on PAP, PVR and systemic vascular resistance, but there was a decrease in resting central venous pressure and PAWP, suggestive of a reduction in venous return related to venodilation [115]. In a subsequent substudy, 18 patients were transitioned from weekly intravenous Levo to daily oral Levo (1–4 mg/day over 8 weeks), and these patients displayed further improvements in 6MWD [116]. Although preliminary, these data are important given the lack of available effective therapy for patients with HFpEF-related PH-LHD. Further adequately powered prospective studies are now required to confirm these promising effects of Levo.

#### 6.2.2. β3-Adrenoreceptor Agonists

Beta-3 adrenergic receptor (β3AR) agonists represent a new class of drugs approved for the clinical treatment of overactive bladder [117]. In the cardiovascular system, β3AR stimulation promotes NO-dependent signaling and indirectly activates cardiac myocyte Na^+^/K^+^-ATPase [118]. Experimental studies indicated that β3AR agonists improved cardiac performance in experimental heart failure and induced the vasodilation of isolated pulmonary vessels from animals and humans [119]. Furthermore, in a porcine model of post-capillary pulmonary hypertension induced by pulmonary vein banding, the oral administration of the β3AR agonist mirabegron resulted in a significant reduction in PVR associated with an improvement in RV performance [119]. These data prompted Garcia-Alvarez et al. to perform an RCT (SPHERE-HF) comparing oral mirabegron (50–200 mg/d) for 16 weeks to placebo in 80 patients with Cpc-PH, mostly in HFpEF (Table 4) [120]. The trial was negative for its primary outcome (no reduction in PVR), but it reported a significant improvement in RV function (secondary outcome). In contrast to these data, Bundgaard et al. reported, in a small, randomized study of 22 patients with advanced HFrEF (both Ipc-PH and Cpc-PH), that a 7-day course of oral mirabegron (300 mg/d) significantly reduced PVR in comparison to placebo [118]. One may, thus, speculate that β3AR agonists could exert some benefits in severe HFrEF, but not in HFpEF, patients. Additional adequately powered studies are required to confirm this hypothesis.

#### 6.2.3. Left Ventricle Assist Device

In advanced HFrEF, a left ventricle assist device (LVAD) is a therapeutic option as a bridge to transplantation, bridge to candidacy or as a destination therapy [121]. Owing to the unloading of the LV, one may expect LVAD implantation to be followed by a significant decrease or normalization in PAP. In a retrospective study of 51 LVAD patients with high PVR before implantation, mPAP and PVR dropped respectively from 43 ± 7 mmHg to 22 ± 6 mmHg and from 6.3 ± 1.2 to 2.2 ± 1.1 WU (*p* < 0.001) upon repeated assessment several months after implantation. These effects persisted up to one year post-transplantation in a subgroup of 14 patients transplanted after LVAD [122]. Anegawa et al. recently reported comparable results in a retrospective study including 89 patients undergoing LVAD surgery, including 56 % with a Cpc-PH profile. All patients normalized PVR within three years [123].

Although these findings support the notion that LV unloading after LVAD implantation may reverse pulmonary remodeling in HFrEF patients with pre-capillary PH, contrasting results were obtained by Al-Kindi et al. In 393 patients with elevated PVR before LVAD implantation, only one third achieved PVR normalization [124]. Furthermore, in a study exploring the diastolic gradient (PAPD-PAWP) as an index of a pre-capillary component of PH in 63 LVAD patients, Imamura et al. found that 43% of patients had a gradient > 5 mm Hg. This implies that a significant number of LVAD patients display persisting PVD. In addition, patients with an elevated diastolic gradient had a significantly worse outcome than patients with a normal gradient [125]. Therefore, implantation of an LVAD may normalize PAP and PVR in a significant number of, but not in all, patients, and those with persisting pre-capillary PH have reduced survival [1]. It may be possible that such patients could benefit from pulmonary vasodilator therapy [126], as suggested by the beneficial effects of sildenafil, a phosphodiesterase-5 inhibitor, in reducing PVR and improving RV function in this population [127,128]. However, current data are not sufficient to make any recommendations [126]. A large prospective study (the SOPRANO study, see below) is currently underway to evaluate the possible benefits of the endothelin receptor inhibitor macitentan in patients with LVAD and persisting elevated PVR.

**Table 4 ijms-24-09971-t004:** Summary of therapeutic clinical trials in PH-LHD.

Study	N, Patients	PVR (WU)	Intervention	Main Outcome	Results
Levosimendan					
Slawsky et al. [112]	146 HFrEF	NA	Levo 6 h, IV	SV, CI, PCWP, RAP, dyspnea score	↗ CI, SV ↘ PCWP, RAP, ↘ dyspnea
Parissis et al. [113]	54 HFrEF	NA	Levo 24 h, IV	RV function, sPAP, Plasma BNP	↗ RV function ↘ sPAP, BNP
Burkhoff et al. [115]	37 HFpEF	3.3 ± 2.6	Levo 6 w, IV	exercise-PCWP 6MWD, PCWP and CVP	↔ PCWP ↗ 6MWD ↘ PCWP, CVP
β3AR agonist					
Bundgaard et al. [118]	22 HFrEF	3.5 ± 2.5	Mirabegron 1 w	PVR, SVR, CI, MAP	↔ MAP, SVR ↗ CI; ↘ PVR
García-Álvarez et al. [120]	80 HFpEF (70%)	4.0 (3.4–4.6)	Mirabegron 16 w	PVR, QOL RV function	↔ PVR, QOL ↗ RV function
ERAs					
Sutsch et al. [129]	36 HFrEF	2.6 ± 1.3	Bosentan 2 w	CO, mPAP, PCWP, RAP	↗ CO; ↘ mPAP, PCWP, RAP
Luscher et al. [130]	157 HFrEF	3.1 ± 0.6	Darusentan 3 w	CI, PCWP, PVR, RAP	↗ CI; ↔ PCWP, PVR, RAP
Anand et al. [131]	642 HFrEF	NA	Darusentan 24 w	LVESV	↔ LVESV
Packer et al. [132]	370 HFrEF	NA	Bosentan 26 w	Death, NYHA, HHF	Early stop (liver toxicity)
Kaluski et al. [133]	94 HFrEF	NA	Bosentan 20 w	sPAP, CI	↔ sPAP, CI ↗ SAEs
Packer et al. [134]	1613 HFrEF	NA	Bosentan 9 m	Death, HF admission	↔ outcome ↗ congestion
Frantz et al. [135]	57 LVAD (Cpc-PH)	4.3 ± 0.9	Macitentan 12 w	PVR, mPAP, RAP, PCWP	↘ PVR ↔ mPAP, RAP, PCWP
Koller et al. [136]	63 HFpEF	3.7 ± 2.5	Bosentan 12 w	6MWD PASP, RAP, TAPSE	↔ all variables Early stop (liver toxicity, ↗ HF)
Vachiery et al. [137]	63 mixed 76% HFpEF (Cpc-PH)	5.6 (3.7–7.3)	Macitentan 12 w	CI, NT-proBNP PCWP, PVR, RAP	↔ PCWP, PVR, RAP, CI; ↗ fluid retention
SERENADE NCT03153111	300 HFrEF (Cpc-PH)	NA	Macitentan 24 w	Plasma NT-proBNP, worsening heart failure	↔ all variables Early stop
PDE5i					
Lewis et al. [138]	34 HFrEF	4.3 ± 0.5	Sildenafil 12 w	Peak VO2, PVR, 6MWD, QOL	↗ VO2, 6MWT, QOL ↘ PVR
Behling et al. [139]	19 HFrEF	NA	Sildenafil 4 w	PAPS, exercise capacity (CPET)	↗ exercise capacity ↘ PAPS
Guazzi et al. [140]	45 HFrEF	NA	Sildenafil 12 m	LVEF, LV diastolic function, exercise capacity, QOL	↗ all variables AE: flushing, headache
Guazzi et al. [141]	32 HFrEF	4.5 ± 0.7	Sildenafil 12 m	Exercise capacity, pulmonary hemodynamics	↗ exercise capacity ↘ PCWP, PVR, mPAP
Amin et al. [142]	106 HFrEF	NA	Sildenafil 12 w	MAP, 6MWD, hospitalization	↔ MAP ↗ 6MWD (ns) ↘ hospitalization (ns)
Cooper et al. [143]	210 HFrEF	NA	Sildenafil 24 w	Symptoms score, 6MWD, QOL and PASP	69 pts analyzed ↔ all variables ↗ temporary withdrawals
Guazzi et al. [144]	44 HFpEF (Cpc-PH+ RV failure)	3.9 ± 1.4	Sildenafil 6–12 m	mPAP, PAWP, PVR, TAPSE	↘ mPAP, PAWP, PVR ↗ TAPSE, ↗ CI
Redfield et al. [145]	216 HFpEF (Ipc-PH)	NA	Sildenafil 24 w	Peak VO2, 6MWD	↔ Peak VO2, 6MWD ↗ AEs (ns)
Andersen et al. [146]	70 HFpEF (Ipc-PH)	2.6 ± 0.9 *	Sildenafil 9 w	PCWP, PAP, CI	↔ PCWP, PAP ↗ CI
Hoendermis et al. [147]	52 HFpEF (65% Ipc-PH)	>3 in 35% pts	Sildenafil 12 w	mPAP, PCWP, CO and peak VO2	↔ all variables
Bermejo et al. [148]	200 LVD (57% Cpc-PH)	3.4 (2.4–4.6)	Sildenafil 24 w, >1 y after valve repair	Composite: death, HF episodes; 6MWD, sPAP, BNP	Clinical worsening ↔ 6MWD, BNP, sPAP
Belyavskiy et al. [149]	50 HFpEF (Cpc-PH)	3.3 ± 0.6	Sildenafil 24 w	6MWD, NYHA, PASP, TAPSE	↗ 6MWD ↘ PASP, NYHA ↗ TAPSE
**SGCs**					
Bonderman et al. [150]	201 HFrEF	3.6 ± 0.3	Riociguat 16 w	mPAP, CI, SVI, PVR	↔ mPAP ↗ CI, SVI; ↘ PVR
Gheorghiade et al. [151]	351 HFrEF	NA	Vericiguat 12 w	Change in NT-proBNP	↔ NT-proBNP
Armstrong et al. [152]	5050 HFrEF	NA	Vericiguat 10.8 m	Composite: CV death, first HHF	↘ primary outcome
Bonderman et al. [153]	39 HFpEF	2.8 ± 1.3	Riociguat 6 h	mPAP, PVR, PCWP, TPG, SV, PAS	↔ mPAP, PVR, PCWP, TPG; ↗ SV, ↘ PAS
Pieske et al. [154]	477 HFpEF	NA	Vericiguat 12 w	NT-proBNP, LA volume, QOL	↔ NT-proBNP, LA volume; ↗ QOL
Udelson et al. [155]	181 HFpEF	NA	Praliciguat 12 w	Peak VO2, 6MWD	↔ peak VO2, 6MWD
Armstrong et al. [156]	789 HFpEF	NA	Vericiguat 24 w	Physical limitation score	↔ score
Dachs et al. [157]	114 HFpEF (Ipc-PH 60%)	3.2 ± 1.7	Riociguat 26 w	CO, PVR, PCWP, TPG, SVR	↗ CO; ↘ PVR, TPG ↔ PCWP, SVR 5 dropouts
Prostacyclin analogs					
Sueta et al. [158]	33 HFrEF	3.6 ± 0.5	Epoprostenol 12 w, IV	6MWD	↗ 6MWD
Califf et al. [159]	471 HFrEF	NA	Epoprostenol 36 w, IV	Mortality, HF symptoms, 6MWD, QOL	Early stop (↗ mortality)
SOUTHPAW NCT03037580	84 HFpEF	NA	Treprostinil 24 w	6MWD, Plasma NT-proBNP, NYHA class	↔ all variables Early stop (slow enrolment)
RECAPTURE NCT04882774	30 HFpEF (Cpc-PH)	NA	Treprostinil	PVR, 6MWD	Not started
Interatrial shunt devices (IASDs)					
Obokata et al. [160]	79 HFpEF	1.5 ± 0.8	IASD up to 6 m	Resting and exercise pulmonary hemodynamics	↘ PVR, Ea ↗ PAC
Shah et al. [161]	626 HFpEF	1.5 (1.1–2.1)	IASD 12–24 m	CV death, HF events QOL	↔ all variables
Pulmonary artery denervation (PADN)					
Zhang et al. [162]	98 mixed 60% HFrEF (Cpc-PH)	6.3 ± 3.2	PADN vs sham procedure 6 m	6MWD, PVR, clinical worsening	↗ 6MWD, ↘ PVR ↘ clinical worsening

For the study population, the PH phenotype (Cpc-PH, Ipc-PH) is shown when specifically mentioned in the study; for PVR, values are means ± SD, or medians (interquartile range) and the symbol * means indexed PVR. Abbreviations: AEs: adverse events; BNP; brain natriuretic peptide; CI: cardiac index; CO: cardiac output; Cpc-PH: combined pre- and post-capillary pulmonary hypertension; CPET: cardio-pulmonary exercise testing; CVD: cardiovascular death; Ea: pulmonary arterial effective elastance; ERAs: endothelin receptor inhibitors; HF: heart failure; HFpEF: heart failure with preserved ejection fraction; HFrEF: heart failure with reduced ejection fraction; HHF: hospitalization for heart failure; IASD: interatrial shunt devices; Ipc-PH: isolated post-capillary pulmonary hypertension; LA: left atrium; LVD: left valve disease; LVESV: left ventricle end-systolic volume; m: months; MAP: mean arterial pressure; mPAP: mean pulmonary artery pressure; ns: no significant; NT-proBNP: N-terminal pro-brain natriuretic peptide; NYHA: New York Heart Association; PADN: pulmonary artery denervation; PCWP: pulmonary capillary wedge pressure; PDE5i: phosphodiesterase 5 inhibitors; PVR: pulmonary vascular resistance; QOL: quality of life; RAP: right atrial pressure; RVEF: right ventricle ejection fraction; SAEs: serious adverse events; SGCs; soluble guanylate cyclase inhibitors; sPAP: systolic pulmonary artery pressure; SV(I): stroke volume (indexed); SVR; systemic vascular resistance; TAPSE: tricuspid annular plane systolic excursion; TPG: transpulmonary gradient; VO2: oxygen consumption; w: weeks; 6MWD: 6 min walk distance; y: year.

### 6.3. Drugs Approved for Pulmonary Artery Hypertension (Table 4)

#### 6.3.1. Endothelin Receptor Antagonists (ERAs)

PH is associated with an increased expression of endothelin-1 in endothelial cells. This high expression is associated with increased PVR, specifically in PAH [163]. ERAs act by blocking endothelin-1 receptors on endothelial vascular smooth muscle, leading to vasodilation, and their use is recommended for the management of PAH [1]. Early animal studies suggested an association between endothelin-1 and the development and progression of heart failure [164,165]. In humans, elevated plasma endothelin-1 correlates with the severity of PH and is associated with poor prognosis in heart failure patients [166,167]. These different results provided the rationale for the use of ERAs in the treatment of PH-LHD.

ERAs in HFrEF

The first clinical study of ERAs in HFrEF patients randomized 36 patients to receive oral bosentan or placebo for 2 weeks and reported a significant reduction in pulmonary and systemic vascular resistance, with an improved cardiac output [129]. In spite of these early positive results, further studies addressing the effects of long-term ERAs using either bosentan [132,133,134] or darusentan [130,131] revealed only a mitigated impact on PVR and PAP. These studies also showed increased congestion and increased heart failure hospitalization with ERA therapy. Furthermore, ERAs were associated with a high incidence of adverse effects, including liver function test alterations, leading to early drug discontinuation in some studies and raising important safety concerns (Table 4). As previously mentioned, a large prospective multicentric RCT (the SOPRANO study, ClinicalTrials.gov Identifier: NCT02554903) addressed the long-term (12 weeks) effects of the ERA macitentan (10 mg/d orally) in 57 HFrEF patients with persisting elevated PVR following LVAD implantation. Preliminary results, published as an abstract, showed a significant reduction in PVR and a safe profile of macitentan [135]. These positive results suggest that LVAD assistance may avoid ERA-related congestion and associated side effects. Final results are awaited before any conclusions can be drawn.

ERAs in HFpEF

ERAs in patients with HFpEF-related PH have been evaluated in two RCTs. The first included 20 patients receiving oral bosentan (125 mg/d) with a total follow up of 24 weeks. The trial was negative in all outcomes (6MWD, PAP and RV function), and the authors reported worsening HF and serious adverse events related to liver toxicity, leading to study interruption after an interim analysis favoring the placebo [136]. The second RCT (MELODY study) addressed the effects of 12 weeks treatment with 10 mg daily macintentan, specifically in patients with Cpc-PH, primarily in HFpEF patients (76% of the cohort). Treatment was not superior to placebo on hemodynamic parameters (PCWP, PVR, RAP) and NT-proBNP levels, and was associated with increased congestion and worsening of NYHA class [137]. The more recent SERENADE trial (ClinicalTrials.gov Identifier: NCT03153111) aimed to assess oral macintentan versus placebo for 52 weeks exclusively in a population of 300 HFpEF patients with Cpc-PH or RV failure. The study was stopped prematurely due to slow enrollment. Preliminary results obtained at 24 weeks follow up (71 patients in each study group) did not show any significant signal in either the primary (reduction in plasma NT-proBNP) or the secondary (worsening heart failure) endpoints (results presented at the 2022 European Society of Cardiology–Heart Failure congress).

Overall, these results indicate that ERAs do not work in patients with PH-LHD related to HFpEF or HFrEF and may be associated with significant side effects, primarily increased congestion and liver toxicity. Possible benefits may nevertheless be observed in patients with severe HFrEF supported by an LVAD, which should still require further investigation.

#### 6.3.2. Phosphodiesterase-5 Inhibitors (PDE5i)

PDE5 is the most abundant phosphodiesterase isoform in the lung. It acts through the degradation of cyclic guanosine monophosphate (cGMP), thereby reducing cGMP-dependent vasodilation [168]. Furthermore, by reducing the effects of cGMP on vascular smooth muscle proliferation and survival, PDE5 may promote pulmonary vascular remodeling [169,170]. PDE5 inhibitors, including sildenafil, tadalafil and vardenafil, are effective in decreasing PAP and pulmonary vascular remodeling, specifically in group 1 PAH [168,169]. Furthermore, PDE5 is upregulated in the hypertrophied RV, and PDE5 inhibition has been associated with improved RV inotropism [171], which might benefit patients with PH related to heart failure who display an intrinsic abnormality of RV contractility.

PDE5i in HFrEF

Several single-center RCTs of limited size (19–106 patients) evaluated the effects of sildenafil (150 mg/d in most studies) administered for 4 weeks to 1 year in patients with PH related to HFrEF [138,139,140,141,142]. Overall, these studies reported beneficial effects of the intervention, both in terms of pulmonary hemodynamics (reduced PAP, PVR and PAWP), cardiac output and exercise capacity, evaluated by the 6MWD. The favorable effects of sildenafil have been confirmed in several meta-analyses [168,172,173], but with limited evidence due to the small sample sizes and heterogeneity of the analyzed studies. Therefore, Cooper et al. recently performed a multicentric RCT (the SilHF trial) to evaluate sildenafil (up to 40 mg three times/day for 6 months) or placebo (2:1 ratio) in a larger sample of HFrEF patients with a systolic PAP > 40 mmHg [143]. The study failed to detect any significant influence on PAP, on exercise capacity or on quality of life. Additionally, there were more temporary withdrawals with sildenafil, and the four deaths in the study were all observed in the sildenafil group (not significant). These disappointing findings, therefore, do not support the use of PDE5 inhibitors in patients with PH in the setting of HFrEF [143].

However, as outlined by the authors, the SilHF study suffered several limitations which may have impacted its conclusions. First, due to slow recruitment, the trial included only 69 patients instead of the planned 210 patients, and therefore may have been underpowered. Second, the SilHF trial included patients with chronic atrial fibrillation (AF), which was an exclusion criterion in several previous studies, and there was a higher incidence of mitral regurgitation (MR) in the sildenafil group. Both AF and MR promote long-standing left atrial overload and reduced atrial compliance, increasing the backward pulsatile flow into the pulmonary circulation, which might partly explain the lack of beneficial clinical or hemodynamic effects of sildenafil in this study [174]. Third, patients were not selected accorded to their PH phenotype (Ipc-PH and Cpc-PH). It would be expected that only patients with a pre-capillary component of PH might benefit from chronic vasodilator therapy, and even more in the presence of significant RV dysfunction [175]. These assumptions are supported by data showing that chronic sildenafil therapy promotes a significant reduction in PVR and improvement in RV function in patients with severe HFrEF awaiting heart transplantation [176,177], as well as in patients with persisting high PVR under LVAD therapy [127,128].

PDE5i in HFpEF

As in the case of HFrEF, studies evaluating PDE5i in patients with HFpEF-related PH yielded conflicting results. In three RCTs totalizing 338 patients allocated to sildenafil (60 mg/d up to 180 mg/d) or placebo for 9–24 weeks, no significant improvement in hemodynamic variables (PCWP, PAP) or exercise capacity (6MWD, peak oxygen consumption) were reported [145,146,147]. Importantly, all these studies included patients in the early stage of PH with an Ipc-PH phenotype and without RV dysfunction. In marked contrast, Guazzi et al. [144] and Belyavskiy et al. [149] assessed sildenafil therapy in two RCTs (44 and 50 patients, respectively), including only HFpEF patients with Cpc-PH and RV failure. These two trials reported significant reductions in PAP and improved RV function and exercise capacity, as well as clinical amelioration (NYHA class). Further adding to the controversial actions of sildenafil in PH-LHD, a study performed on 200 patients with persistent PH (56% Cpc-PH, preserved EF in 56%) secondary to valvular disease evaluated the effects of chronic sildenafil, administered for 6 months, starting at least 1 year after valve repair [148]. The results were unequivocal, showing no hemodynamic or functional improvement, but showing significant clinical worsening in the sildenafil group [148].

Overall, current evidence does not support the general use of sildenafil for PH-LHD, either related to HFrEF and HFpEF. This statement is particularly true for patients with Ipc-PH, no RV dysfunction and chronic AF. Furthermore, sildenafil is associated with significant clinical worsening in patients with persistent PH following valve repair. It remains possible that sildenafil may exert some positive effects in patients with Cpc-PH and RV dysfunction (both in HFrEF and HFpEF), and in patients with persisting high PVR undergoing LVAD therapy, but additional studies are required to confirm such possibilities. In the meantime, it seems inappropriate to use PDE5i in PH-LHD patients outside of clinical trials.

#### 6.3.3. Soluble Guanylate Cyclase Stimulators (SGCSs)

SGC stimulators, including the drugs riociguat, vericiguat and praliciguat, increase the sensitivity of soluble guanylate cyclase (sGC) to nitric oxide (NO) [178], thereby enhancing the formation of cyclic GMP, the major second messenger of NO signaling [179]. This results in vasodilation and antiproliferative, anti-inflammatory and antifibrotic effects [178]. Furthermore, these compounds may also directly activate SGC independently from NO. This is of particular interest when the availability of NO is reduced as a consequence of endothelial dysfunction (reduced NO biosynthesis) or oxidative stress (direct NO inactivation), which prevail in the context of heart failure [178]. The SGC riociguat is approved for the therapy of PAH and CTPEH (chronic thromboembolic PH) [1]. Decreased NO bioavailability and reduced cGMP activity are observed in heart failure [180] and contribute to disease progression, providing a strong rationale for the use of SGCs in the context of heart failure [178].

SGCSs in HFrEF

Three key RCTs addressed the effects of SGCs in HFrEF patients. Bonderman et al. (LEPHT study) compared the hemodynamic effects of three groups of an incremental regimen of riociguat (1.5–6 mg/d) against placebo administered for 16 weeks [150]. At the highest dose, riociguat did not change mPAP (primary outcome), but significantly decreased the PVR and increased the cardiac index without changes in heart rate or systemic blood pressure. This supported the finding that riociguat induced pulmonary vasodilation with a secondary increase in cardiac output. The second trial was conducted by Gheorghiade et al., who compared vericiguat (one of four daily doses of 1.25–10 mg) to placebo in 351 HFrEF patients within 4 weeks of a worsening chronic HF event (SOCRATES-REDUCED study) [151]. Although vericiguat did not result in a significant change in the primary outcome (plasma NT-proBNP), secondary analyses suggested a dose-response effect with a greater reduction in NT-proBNP with higher doses of vericiguat.

The recent VICTORIA study is the largest RCT to date on SGCs in HFrEF patients. The trial evaluated vericiguat (10 mg/d) vs placebo over a median of 10.8 months in 5050 HFrEF patients with a recent worsening of HF symptoms [152]. The drug was well tolerated and allowed a significant reduction in the primary composite outcome of death from cardiovascular causes or first hospitalization for heart failure, primarily driven by a lower incidence of first HF hospitalization. These results are of major importance, given the prognostic benefit obtained with vericiguat over the standard of care. Therefore, recent guidelines recommend considering vericiguat in selected patients with HFrEF and a recent worsening of HF (grade 2a recommendation) [93].

SGCSs in HFpEF

SGCs have been assessed in several RCTs in HFpEF patients, as recently extensively reviewed by Liang et al. [181]. In one hemodynamic study evaluating the acute (6 h) effects of vericiguat (0.5–2 mg orally) in 39 patients, Bonderman et al. did not find any significant effects of the intervention on mPAP, PAWP and PVR [153]. Three large RCTs addressed the long-term effects of SGCs on clinical endpoints. In the SOCRATES-PRESERVED study (*n* = 477), vericiguat up to 10 mg/day for 12 weeks did not change NT-proBNP levels or left atrial volume, but improved quality of life [154]. In the VITALITY-HFpEF study (*n* = 789), vericiguat up to 15 mg/day for 24 weeks did not improve physical limitation, as assessed by the Kansas City Cardiomyopathy Questionnaire [156]. Udelson et al. evaluated praliciguat (40 mg/d for 12 weeks) in 181 patients and did not find any improvement in exercise capacity, as measured by peak VO2 and 6MWD [155].

In a recent phase II multicentric RCT with both hemodynamic and clinical interest, Dachs et al. compared riociguat (up to 4.5 mg/d) to placebo in 114 patients with PH related to HFpEF [157]. Treatment significantly improved resting cardiac output by an average of 0.54 L (primary outcome), and it significantly decreased PVR while not changing PAWP, as assessed by right heart catheterization. In the absence of change in heart rate and systemic vascular resistance, the authors attributed the change in CO as reflecting an increased stroke volume following pulmonary vasodilation. Notwithstanding these positive hemodynamic effects, there were no observed changes in secondary outcome measures, including plasma NT-proBNP, exercise capacity and quality of life. Furthermore, there were significantly more adverse events resulting in more dropouts from the study with riociguat. An interesting observation of the study was that in 18 patients with a Cpc-PH phenotype, a greater improvement in hemodynamic parameters in response to riociguat was noted, suggesting a possible benefit of the drug in pulmonary vascular remodeling. As stated by the authors, this exploratory result should prompt future trials to focus on Cpc-PH patients to clarify the therapeutic potential of riociguat in this subgroup [157].

To sum up current knowledge, it appears that riociguat and vericiguat are generally well tolerated and provide favorable hemodynamic effects in both HFrEF and HFpEF patients, with a significant increase in cardiac output and reduction in PVR. Additional studies specifically exploring these effects in patients with a Cpc-PH profile should be conducted. Furthermore, vericiguat significantly improves clinical outcomes in HFrEF patients with a recent worsening of HF symptoms and is therefore recommended in the management of such patients. In contrast, clinical benefits from SGCs in HFpEF remain not clearly defined, as indicated by an improved quality of life reported with vericiguat, contrasting with no such effects found with riociguat and praliciguat. Therefore, additional studies are needed in this specific category of HF patients.

#### 6.3.4. Prostacyclin Analogs

Prostacyclin (prostaglandin I2, PGI2) is a prostanoid produced via PGI2 synthase in vascular endothelial cells which acts on specific receptors in platelet and endothelial cells. It promotes potent vasodilator effects decreasing both pulmonary and systemic vascular resistance, reduces platelet aggregation and displays antiproliferative actions [182]. Various prostacyclin analogs, including beraprost, treprostinil, iloprost and epoprostenol, are recommended in patients with PAH [1]. A few studies have evaluated the role of these therapeutic approaches in patients with PH-LHD.

Prostacyclin analogs in HFrEF

The pulmonary vasodilatory effect of prostacyclin analogs has been well documented in patients with severe HFrEF and elevated PVR. In such patients, the inhalation of iloprost produced an acute reduction in PVR and an increase in cardiac output, with only moderate systemic effects [183,184]. Two longer-term studies evaluated the potential clinical benefits from a continuous infusion of epoprostenol. The first small trial (33 patients) reported an improvement in the 6MWT following 12 weeks of treatment [158]. However, such results were not confirmed in a later study evaluating intravenous epoprostenol for 36 weeks [159]. The study was prematurely discontinued due to increased mortality rates (mostly related to worsening of progressive congestive heart failure) and no evidence of improved quality of life, despite a significant increase in cardiac index and decrease in PCWCP [158,159]. These negative results indicate that prostacyclin analogues should not be used in PH related to HFrEF outside of controlled studies.

Prostacyclin analogs in HFpEF

In an experimental study evaluating two animal models of metabolic syndrome associated with HFpEF and PH, treatment with treprostinil for 16 weeks reduced pulmonary pressure, improved right heart function and blunted hyperglycemia [185]. These data support the hypothesis that chronic trepostinil therapy may improve pulmonary endothelial dysfunction related to metabolic syndrome, which is an important pathophysiological mechanism of PH in the setting of HFpEF. Such hypothesis has been evaluated in an RCT comparing oral treprostinil (0.125 up to 18 mg/d) to placebo for 24 weeks in HFpEF patients with PH (Southpaw study, NCT03037580). Unfortunately, the trial was terminated early because of slow enrollment, and an analysis of the first 52 patients did not show any impact of treatment on 6MWD or NT-proBNP levels. A novel study (non-randomized), currently in the planning stage, aims to evaluate oral treprostinil in patients with HFpEF with Cpc-PH using the CardioMEMS^TM^ monitoring tool (NCT04882774).

In summary, only limited information currently exists regarding prostacyclin analogs for the treatment of PH-LHD. Long-term epoprostenol has been associated with detrimental effects in HFrEF. In HFpEF, some encouraging experimental data have been produced using treprostinil, but no clinical data have so far confirmed these promising effects.

### 6.4. Other Therapeutic Options

#### 6.4.1. Interatrial Shunt Devices

A recently developed interatrial shunt device (IASD) able to reduce left atrial pressure via left atrial decompression is currently being investigated in the treatment of heart failure [186]. A pooled analysis of two studies of 79 patients with HFpEF revealed that the IASD was associated with reduced PVR and increased PA compliance, both at rest and during supine exercise, up to 6 months after insertion. These data prompted a prospective RCT (REDUCE LAP-HF II) including 626 HFpEF patients with exercise PAWP > 25 mmHg and PVR < 3.5 WU, treated either with an IASD or a sham procedure for up to 24 months (Table 4) [161]. The study did not demonstrate any significant effect of the device on the primary composite outcome of cardiovascular death, non-fatal ischemic stroke and rate of total heart failure events. Given that the atrial shunt promotes an increase in pulmonary blood flow, this approach might be counterproductive in patients with pulmonary vascular remodeling, whereas it might be helpful in patients with isolated post-capillary PH via the reduction in PAWP [187]. To address this hypothesis, Borlaug et al. performed a secondary analysis of the previous trial, discriminating patients with or without latent PVD, according to a peak exercise PVR value > 1.74 WU [187]. The analysis revealed that the IASD was associated with worse outcomes in the presence of latent PVD, contrasting with a signal of clinical benefit in the absence of PVD. These results indicate that HFpEF patients with Ipc-PH and no evidence of latent PVD, as assessed during exercise hemodynamics, may benefit from shunt-mediated left atrial unloading. This hypothesis should be explored in future clinical trials evaluating the IASD.

#### 6.4.2. Pulmonary Artery Denervation

Increased activity of the sympathetic nervous system has been demonstrated in patients with PAH, as evidenced by elevated plasma catecholamines and heightened muscle sympathetic nerve activity [188]. Furthermore, a local pulmo-pulmonary baroreflex elicited by stretch receptors in the pulmonary artery has been shown to promote local sympathetic activation. Elevated systemic and local sympathetic activity elicit pulmonary vasoconstriction via α-1 receptors and may greatly contribute to pulmonary vascular remodeling, providing a strong rationale for therapies interfering with the sympathetic nervous system in PAH. An extensive review on this topic has been recently published [189]. In experimental animal studies, the percutaneous sympathetic denervation of the proximal pulmonary arteries resulted in a significant reduction in PVR, PAP and vascular remodeling [190,191]. On the basis of these results, Chen et al. published the first human study (PADN-1 study) evaluating PADN, performed by transcatheter radiofrequency ablation at the level of the main pulmonary artery, in 13 PAH patients (Table 4). At 3 months follow up, the procedure resulted in significant reduction in mPAP, and significant improvement in the 6MWT and in RV function [192]. These beneficial effects have been confirmed in a prospective RCT (PADN-CFDA trial) comparing PADN vs sham procedure in 128 PAH patients [193]. The positive impact of PADN in PAH has been further highlighted in a recent meta-analysis on this procedure [194].

Owing to the major significance of increased sympathetic activity in patients with heart failure and given the importance of progressive pulmonary vasoconstriction and remodeling leading to the phenotype of Cpc-PH, the PADN procedure might be of particular interest in this category of patients. In phase II of the PADN-1 study, 66 patients with PH of different etiologies underwent PADN. Reductions in PAP and PVR and improved functional status (6MWD) occurred at 6 months across all PH etiologies, including 18 patients with PH-LHD [195]. Furthermore, in an animal model of PH-LHD induced by aortic banding, PADN significantly improved PA remodeling, hemodynamics and RV function [196]. To extend these findings, Zhang et al. randomized 98 patients with Cpc-PH (40% HFpEF, 60% HFrEF) to PADN or sham PADN + sildenafil. After 6 months, patients treated with PADN displayed a significant increase in 6MWD, a lower PAP and PVR and a reduction in clinical worsening. Further RCTs are needed to confirm these encouraging results and to address long-term clinical outcomes following PADN therapy.

In summary, there is growing evidence that PADN permits significant hemodynamic and clinical benefits in patients with PH of various etiologies, including patients with a Cpc-PH profile in the context of heart failure. Although more studies on this issue are needed, these promising results suggest that PADN might soon find a place in the management of PAH in general, and in PH-LHD in particular.

## 7. Conclusions and Future Perspectives

Pulmonary hypertension related to left heart disease represents the most frequent form of PH, which will continue to pose significant challenges in terms of public health and economic burden in years to come. According to the most recent statistics from the American Heart Association and National Institute of Health, the crude prevalence of cardiovascular diseases has increased by 29.5% from 2007 to 2017 worldwide and will account for >22.2 million deaths by 2030 [197]. On a pathophysiological standpoint, the triggering event for PH in LHD is the passive upstream transmission of elevated left-sided pressures and rising diastolic, systolic and mean pulmonary pressures. The consecutive reduction in pulmonary artery compliance increases the pulsatile hydraulic load imposed on the right ventricle. At this stage, treatment should aim to reduce congestion and left-sided filling pressures, primarily with the judicious use of diuretics and the appropriate adjustment of guideline-directed medical therapies. The optimization of such therapies may be facilitated via the remote monitoring of pulmonary artery pressure using an implanted sensor.

The control of left-sided filling pressure is crucial to prevent or limit the risk of further development of pulmonary vascular disease promoted by capillary stress failure induced by the chronic increase of capillary wall tension. This mechanical insult, in turn, triggers a cascade of molecular events fostering endothelial dysfunction, pulmonary vasoconstriction and inflammatory-driven remodeling of pulmonary vessels with the progressive thickening of venous, capillary and arterial walls, narrowing the vessel lumen. Several genetic and metabolic factors, notably, diabetes and obesity, are highly prevalent in the context of heart failure with preserved ejection fraction and are key favorizing factors for such remodeling. The corollary is an increase in pulmonary vascular resistance, determining a pre-capillary phenotype of PH. These changes result in an increase in the steady hydraulic load further charging the right ventricle, up to an unstainable point that marks the onset of right ventricle uncoupling and failure, with negative consequences. Intrinsic abnormalities in the RV myocardium in the context of heart disease, reducing its contractile reserve, contribute to precipitating this dismal evolution.

Can we influence the course of PH-LHD at the stage of increased PVR? The normalization of PVR in a significant proportion of patients supported by a left ventricle assist device for terminal heart failure is encouraging in this context. It indicates that some form of reverse remodeling may occur in the pulmonary circulation, and hence that PVD may not be irreversible. Unfortunately, all targeted therapies aiming to reduce PVR, which proved useful in other forms of pre-capillary PH, have mostly failed to show significant positive effects in patients with PH-LHD. A noticeable exception is the clinical benefit recently demonstrated with vericiguat in patients with advanced heart failure (VICTORIA study). A potential drawback of pulmonary vasodilators may be the increase in left-sided filling pressures secondary to the increase in pulmonary blood flow. The increased congestion noted in some trials with pulmonary vasodilators indeed supports such assertion. However, we should not be discouraged in the face of these disappointing results, as many unresolved questions remain to be answered. Key issues in future trials will be the proper selection of patients, according to the nature of heart failure (preserved or reduced ejection fraction), the phenotype of PH (Ipc-PH vs Cpc-PH) and the degree of RV dysfunction, as well as the clear definition of outcome variables (hemodynamic, clinical). Additionally, studies will need to address the role of pulmonary vasodilators in the context of new heart failure therapies, including angiotensin receptor–neprilysin inhibitors and SGLT2 inhibitors.

Some novel strategies may prove beneficial to patients with PH-LHD. As an example, the recent observation of clinical improvement in patients treated with oral Levosimendan is an encouraging development. Positive results obtained with pulmonary artery denervation are also of considerable interest, and this approach could represent a breakthrough in years to come. Finally, the improved understanding of the molecular mechanisms of pulmonary vascular remodeling may permit us to envision, with reasonable optimism, novel specific targeted therapies for the future.

## Figures and Tables

**Figure 1 ijms-24-09971-f001:**
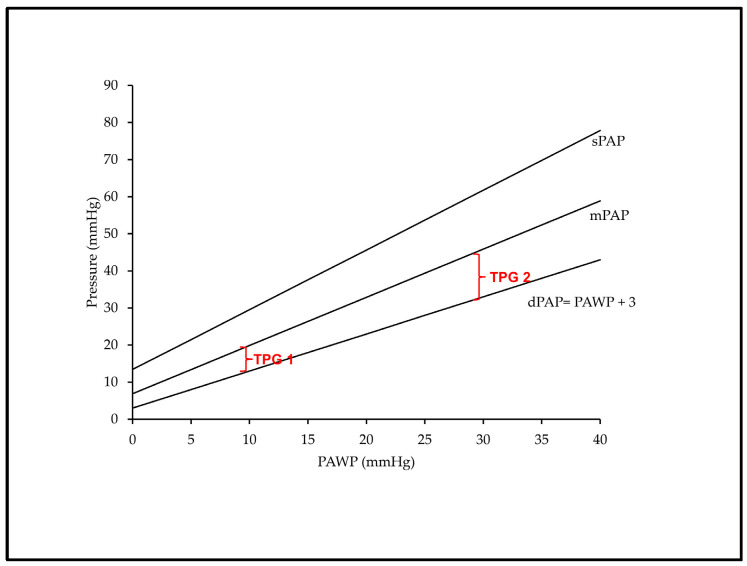
Schematic representation of the impact of PAWP on dPAP, mPAP, sPAP and TPG at a given stroke volume of 80 mL. Backwards transmission of PAWP to dPAP at a 1:1 mmHg ratio and >1:1 mmHg rise in sPAP and mPAP, resulting in a TPG of 7 mmHg if PAWP is at 10 mmHg (TPG 1) and of 17 mmHg if PAWP is at 30 mmHg (TPG 2). See text for abbreviations. Adapted from Ref. [30].

**Figure 2 ijms-24-09971-f002:**
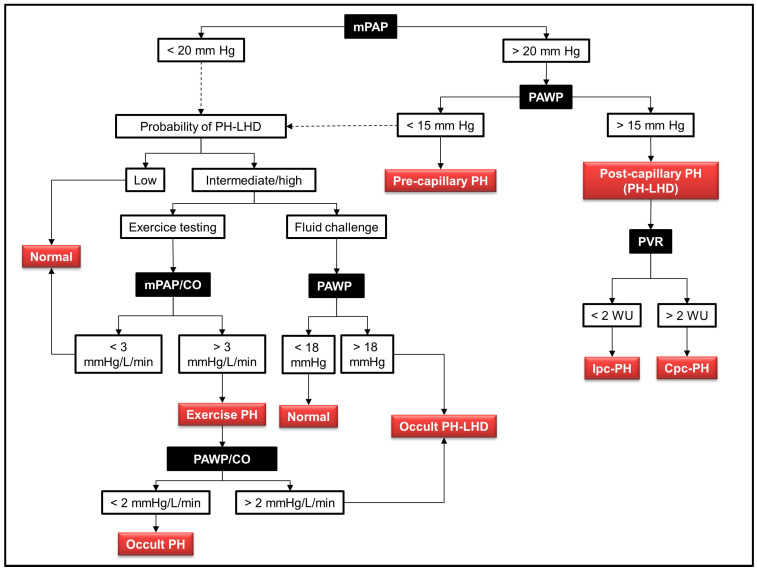
Algorithm for the hemodynamic diagnosis of pulmonary hypertension associated with left heart disease (PH-LHD). The black squares indicate the hemodynamic variables obtained during right heart catheterization. The red squares indicate the different diagnoses according to the measured hemodynamic values. The dashed lines indicate the procedure to follow in case of normal hemodynamic variables: in the presence of a clinical probability of PH-LHD, provocative tests are warranted, including either exercise testing or fluid challenge. Abbreviations: PH: pulmonary hypertension; mPAP: mean pulmonary artery pressure; PAWP: pulmonary artery wedge pressure; CO: cardiac output; Ipc-PH: isolated post-capillary pulmonary hypertension; Cpc-PH: combined pre- and post-capillary pulmonary hypertension.

**Table 1 ijms-24-09971-t001:** Hemodynamic definition of PH-LHD.

Variable	N	PAH	Ipc-PH	Cpc-PH
mPAP (mmHg)	<20	>20	>20	>20
PAWP (mmHg)	<15	<15	>15	>15
PVR (mmHg/L/min = Wood Units)	<2	>2	<2	>2

Cpc-PH: combined pre- and post-capillary pulmonary hypertension; Ipc-PH: isolated post-capillary pulmonary hypertension; mPAP: mean pulmonary artery pressure; PAH: pulmonary arterial hypertension; PAWP: pulmonary artery wedge pressure; PVR: pulmonary vascular resistance. Table prepared according to data published in Ref. [1].

**Table 2 ijms-24-09971-t002:** Hemodynamic definition of PH-LHD upon provocative testing.

Variable (Provocative Testing)	N	PH-LHD
1. Exercise		
mPAP/CO (mm Hg/L/min)	<3	>3
PAWP/CO (mm Hg/L/min)	<2	>2
2. Volume challenge		
PAWP at the end of volume challenge (<5 min)	<18	>18

CO: cardiac output; mPAP: mean pulmonary artery pressure; PAWP: pulmonary artery wedge pressure. Table prepared according to data published in Refs. [1,42,55].

**Table 3 ijms-24-09971-t003:** Right ventricle to pulmonary artery (RV-PA) coupling.

Variable	Formula (Units)
Ees (RV contractility)	ESP/ESV (mmHg/mL)
PVR (RV static afterload)	(mPAP-PAWP)/CO (mmHg/L/min, N < 2)
PAC (RV pulsatile afterload)	SV/(PAPS-PAPD) (mL/mmHg, N > 2.3)
Ea (global RV afterload)	ESP/SV (mmHg/mL)
ESP	(1.65 × mPAP) − 7.79 (mmHg)
*RV-PA coupling*	
Calculation (units)	Cut-off values for RV-PA uncoupling
Ees/Ea (dimensionless)	<0.8
SV/ESV (%)	<0.54
(1 − EF)/EF (%)	EF < 35%
TAPSE/PAPS (mm/mm Hg)	<0.31

Ea: pulmonary artery effective elastance; Ees: end-systolic elastance; ESP: end-systolic pressure; ESV: end-systolic volume; SV: stroke volume; mPAP, PAPD, PAPS: mean, diastolic and systolic pulmonary artery pressure; PAWP: pulmonary artery wedge pressure; EF: ejection fraction; TAPSE: tricuspid annular plane systolic excursion; PAC: pulmonary artery compliance; CO: cardiac output. Table prepared according to data published in Refs. [64,75,76].

## Data Availability

Not applicable.

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
