# Peer review of "Pulmonary Hypertension in Left Heart Diseases: Pathophysiology, Hemodynamic Assessment and Therapeutic Management"

_ijms, 2023, doi:10.3390/ijms24129971_

Round 1

Reviewer 1 Report

Congratulation for your hard work! It is an interesting and valuable paper!

The article regarding Novel Therapeutic Targets for Pulmonary Arterial Hypertension is: 

- a very strong article with a very good introduction

- has strong statistical data with valuable and new references

- has very good images

- it has an interesting approach of the subject making it valuable to fellow colleagues to better understand this disease

Author Response

We are thankful to the reviewer for his/her positive comments on our article.

Reviewer 2 Report

I have read the paper with great interest. This is a very comprehensive state of the art for the pathophysiology and treatment of  PH-LHD.

I have only some minor technical suggestions:

·        a table of contents seems to be useful,

·        abbreviations are used very often - a list of them would help very much with reading,

·        long sentences expanding for more than three lines should be avoided

Author Response

We are thankful to the reviewer for his/her positive comments.

Responses to specific comments:

1) We have inserted a table of contents (page 2 of the revised manuscript)

2) We have inserted a complete list of abbreviations (Page 1 of the revised mansucript)

3) We have shortened several long sentences throughout the revised mansucript  

Reviewer 3 Report

The review by Ltaief et al is complete and the topic is adequately investigated.

Author Response

(The authors gave the same response as above.)
